# Spectral and Redox Properties of a Recombinant Mouse Cytochrome *b*561 Protein Suggest Transmembrane Electron Transfer Function

**DOI:** 10.3390/molecules28052261

**Published:** 2023-02-28

**Authors:** Alajos Bérczi, Zsuzsanna Márton, Krisztina Laskay, András Tóth, Gábor Rákhely, Ágnes Duzs, Krisztina Sebők-Nagy, Tibor Páli, László Zimányi

**Affiliations:** 1Institute of Biophysics, Biological Research Centre Szeged, H-6726 Szeged, Hungary; 2Department of Biotechnology, University of Szeged, H-6726 Szeged, Hungary

**Keywords:** ascorbate, CD spectroscopy, cytochrome *b*561 protein, electron transfer, EPR spectroscopy, homology modelling, *Mus musculus*, optical spectroscopy, putative tumor suppression

## Abstract

Cytochrome *b*561 proteins (CYB561s) are integral membrane proteins with six trans-membrane domains, two heme-*b* redox centers, one on each side of the host membrane. The major characteristics of these proteins are their ascorbate reducibility and trans-membrane electron transferring capability. More than one CYB561 can be found in a wide range of animal and plant phyla and they are localized in membranes different from the membranes participating in bioenergization. Two homologous proteins, both in humans and rodents, are thought to participate—via yet unidentified way—in cancer pathology. The recombinant forms of the human tumor suppressor 101F6 protein (Hs_CYB561D2) and its mouse ortholog (Mm_CYB561D2) have already been studied in some detail. However, nothing has yet been published about the physical-chemical properties of their homologues (Hs_CYB561D1 in humans and Mm_CYB561D1 in mice). In this paper we present optical, redox and structural properties of the recombinant Mm_CYB561D1, obtained based on various spectroscopic methods and homology modeling. The results are discussed in comparison to similar properties of the other members of the CYB561 protein family.

## 1. Introduction

Cytochrome *b*561 proteins (CYB561s) are integral membrane proteins with six trans-membrane domains, two heme-*b* redox centers, and trans-membrane electron transporting properties [1,2]. These proteins came into the focus of interest about 25 years ago when it was shown that more than one CYB561 can be found in a wide range of animal and plant phyla. The CYB561s are localized in membranes different from the membranes participating in bioenergization. The two heme-*b* chromophores, one on each side of the membrane housing the CYB561 in question, are coordinated by four highly conserved His residues localized in the central four trans-membrane helices [3]. The major characteristics of these proteins are their ascorbate (ASC) reducibility and trans-membrane electron transferring capability [4]. CYB561s have been classified into seven groups based on primary structural similarities [1]. Five out of the seven groups contain CYB561s with only the core six trans-membrane domains [2] that show significant similarity to the structure of the bovine adrenal gland chromaffin granule cytochrome *b*561 protein (Bt_CYB561A1). This latter protein was the first and is the denominator member of the CYB561 protein family [1]. In spite of the rather intense studies in the past two decades, the atomic structure has been resolved for only two members of the CYB561 protein family, one from *Arabidopsis thaliana* (At_CYB561B2) and the human duodenal protein (Hs_CYB561A2) [5,6].

Two CYB561s that might be involved in cancerous phenomena have been experimentally characterized so far. The recombinant forms of the human tumor suppressor 101F6 protein (Hs_CYB561D2 or human Cyb561d2 protein) and its mouse ortholog (Mm_CYB561D2 or mouse Cyb561d2 protein) have been expressed in yeast and their optical and redox properties established [7,8,9,10,11]. Recently, the ferric reductase activity of detergent purified and lipid nanodisc embedded Hs_CYB561D2 was directly demonstrated [12]. Tsubaki et al. [1] predicted the presence of another homologue CYB561 protein both in humans (Hs_CYB561D1; known also as human Cyb561d1 protein) and in mouse (Mm_CYB561D1 or mouse Cyb561d1 protein), however very little is known about these two proteins. The primary structure of both the human and the mouse protein has 229 amino acids and sequence alignment shows 90% identity and 95% similarity between the two sequences. It is assumed that they are also involved in tumor suppression and have physical-chemical characteristics very similar to those of the CYB561D2s, however, a detailed study has not yet been published about these proteins. The human Cyb561d1 gene product has been detected in many tissues (https://www.proteinatlas.org/ENSG00000174151-CYB561D1/tissue, accessed on 27 February 2023) and the gene activity was evident in a wide variety of biological processes, such as in different tumorous processes [13,14], in blocking mitosis of both U20S and HeLa cells [15], in upregulation of expression of Retinoid X receptors in pancreatic β-cells [16], in aging [17], in type 2 diabetes [18] and in cognitive function [19]. The expression of the mouse Cyb561d1 gene was highest in thymus, spleen, colon, and large intestine (https://www.ncbi.nlm.nih.gov/gene/72023, accessed on 27 February 2023).

To obtain a deeper insight into the possible biological function of these proteins, we expressed the recombinant form of the mouse protein in yeast (*Saccharomyces cerevisiae*) cells, purified partially from the yeast membranes, and determined the basic optical, redox and electron paramagnetic properties of Mm_CYB561D1. No electronic interaction between the two heme-*b* centers was observed by circular dichroism spectroscopy. The binding constants of the two putative ascorbate binding sites and the midpoint reduction potentials of the two heme-*b* centers were comparable to other members of the CYB561 protein family. No spectral differences were discernable for the two hemes, in contrast to several other CYB561 proteins. The redox titration experiments were analyzed in the framework of a complex model and the consequences of this model are discussed. The two hemes showed a highly asymmetric low-spin (HALS) character. We have also generated the putative 3D structure of the protein by homology modelling, and we discuss the likely transmembrane electron transfer pathways.

## 2. Results

Since more than 50% of the amino acids of Mm_CYB561D1 are predicted to be located in the transmembrane domains in a hydrophobic milieu, some well-known and frequently used nonionic detergents with c.m.c. value less than 0.5 mM were tested for solubilization efficacy. As shown in Appendix A the most efficient solubilizing agent proved to be the dodecyl-β-D-maltoside (DDM). The amount of ascorbate-reducible Mm_CYB561D1 in the solubilized fractions was determined from the ascorbate-reduced minus ferricianide-oxidized difference spectrum of the solubilized fractions. Although C12E8 and SB3-14 as well as SMA(2:1) polyelectrolite—which has recently been used in many cases for solubilizing integral membrane proteins [20,21,22,23,24]—solubilized about the same amount of proteins as DDM did, the specific content of ascorbate-reducible Mm_CYB561D1 in the solubilized fractions was much lower than that in the DDM-solubilized fraction. Thus, for affinity chromatography, only DDM-solubilized membrane fractions were collected and further processed.

The Mm_CYB561D1 could not be purified to homogeneity by His_6_-tag affinity chromatography, thus the final sample we used in our spectral and electrochemical characterization contained partially purified Mm_CYB561D1. Since the aromatic amino acid content of Mm_CYB561D1 is comparable to that of other CYB561 proteins [1], only the presence of contaminating proteins could explain the rather high absorbance at 280 nm in the affinity-purified final fraction (Figure 1A) and it was the reason that the A(280 nm)-to-A(Soret peak) ratio—that is below 0.4 for other, highly purified CYB561 proteins [25,26]—was higher than 1.1 in our final samples. Since the yeast membranes we used do not contain ascorbate-reducible cytochromes, only a minor amount of dithionite-reducible ones (Figure 1B), it is safe to assume that protein contamination did not influence the results obtained by the ascorbate and redox titration of the His_6_-tagged, affinity-purified, recombinant Mm_CYB561D1.

Figure 2 shows the circular dichroism (CD) spectra of oxidized and reduced Mm_CYB561D1 in 50 mM phosphate buffer (pH 7.0) in the presence of 0.5 mM DDM between 380 and 600 nm.

Except for a minor sign in the Soret-band of the dithionite-reduced spectrum, no exciton splitting can be seen in either band, that would indicate electronic interaction between the two heme-*b* chromophores either in the oxidized or the reduced state.

As it was predicted by Tsubaki et al. [1], the Mm_CYB561D1 was an ascorbate reducible *b*-type cytochrome. When partially purified Mm_CYB561D1 was reduced by increasing the concentration of ascorbate at pH 7 (Figure 3A), the pattern was similar to that obtained for other CYB561 proteins earlier [27,28,29]. Difference spectra obtained by subtracting the fully oxidized spectrum from all others resulted in the spectral matrix **D**, shown in the Appendix A. Singular value decomposition (SVD) (Equation (9), see Materials and Methods) yielded two significant eigenvector pairs, i.e., matrix **D** had a rank of 2. The significant titration eigenvectors and their successful fit by Equation (10) (Materials and Methods) is plotted in Figure 3B and in 2D plot in Appendix A. The two characteristic ascorbate concentration values for the Mm_CYB561D1 were around K_1_ = 0.045 ± 0.007 mM and K_2_ = 2.34 ± 0.50 mM. These two values (a) may be interpreted as the ascorbate concentrations where half of either of the two heme-*b* centers in the protein is already reduced and (b) are close to the values obtained for other CYB561 proteins [4].

A calculation based on Equation (11) (Materials and Methods) revealed the presence of two major components with identical Soret and rather similar α-bands (Figure 3C). The oxidized (black), high ASC affinity (low ASC concentration) reduced (red) and low ASC affinity (high ASC concentration) reduced (yellow) spectra in Figure 3C correspond to the black, red and yellow squares in Appendix A. The yellow spectrum contains a minor oxidized contribution, and the subtraction of this oxidized “contamination” resulted in the fully reduced blue spectrum. The near identity of the two spectra is at variance with the results obtained for the purified, recombinant, His_6_-tagged At_CYB561A1 [30] and Mm_CYB561D2 [11] proteins where the calculated spectra for the low and high ASC concentration-reduced hemes were distinctly different in the α-band.

Redox titration of the fully oxidized Mm_CYB561D1 by dithionite under anaerobic conditions yielded the set of absorption spectra in Figure 4A. The difference spectra in matrix **D**, obtained from this set (not shown), were analyzed based on the reaction scheme depicted in Figure 4B and the Nernst equation as follows.
**D** = **P C**^T^ = **U S V**^T^ = **U V**_S_^T^ = (**U R**) (**R**^−1^
**V**_S_^T^) (1)
where **P**_mxr_ and **C**_nxr_ are the spectra and potential dependent amounts of the species titrated in two steps (see Figure 4B), **U**_mxr_ and **V**_nxr_ are the SVD eigenvectors, **S**_rxr_ is the diagonal matrix of the singular values, **V**_S_ = **V S**^T^, and **R**_rxr_ is the transformation connecting the physical and abstract spectra and the physical and abstract titration patterns. The rank, r, in the present experiments is 2.

The relative amounts for the four species shown in the scheme in Figure 4B are obtained as functions of the potential E, in mV, as follows (N is the number of electrons transferred in each elementary step, assumed to be 1):(2)$$\begin{array}{c} c_{\mathrm{red}/\mathrm{ox}}=c_{\mathrm{ox}/\mathrm{ox}\;}10^{(E_{m,1}-E)\frac{N}{60}}\\ c_{\mathrm{ox}/\mathrm{red}}={\alpha \;c}_{\mathrm{red}/\mathrm{ox}\;},\;\mathrm{with}\;\alpha =10^{(E_{m,3}-E_{m,1})\frac{N}{60}}\\ c_{\mathrm{red}/\mathrm{red}}=c_{\mathrm{ox}/\mathrm{ox}\;}10^{(E_{m,1}+E_{m,2}-2E)\frac{N}{60}}\\ c_{\mathrm{ox}/\mathrm{ox}}=1/\left[1+\left(1+\alpha \right)10^{(E_{m,1}-E)\frac{N}{60}}+10^{(E_{m,1}+E_{m,2}-2E)\frac{N}{60}}\right] \end{array}$$

From Equation (1) it follows for the titration eigenvectors:**V**_S_ = **C R**^T^(3)
where **C** = [(c_red/ox_ + c_ox/red_),c_red/red_]

Null model:

With the arbitrary choice α = 0 one assumes that only the upper branch of the scheme in Figure 4B is operational, meaning that the second heme can only be reduced after the first one. Strictly speaking, this is only possible if the reduction potential difference between the two heme-*b* centers is infinite. In this case the concentration matrix used for fitting the V eigenvectors is
(4)$$C=\left[10^{(E_{m,1}-E)\frac{N}{60}}\;\;,\;\;10^{(E_{m,1}+E_{m,2}-2E)\frac{N}{60}}\right]/\left(1+10^{(E_{m,1}-E)\frac{N}{60}}+10^{(E_{m,1}+E_{m,2}-2E)\frac{N}{60}}\right)$$

Simultaneous nonlinear least squares fit of the **V**_S_ matrix according to Equation (3) provides N, ^0^E_m,1_, ^0^E_m,2_ and the elements of **R**. The result of this fit is shown as lines in Figure 4B, together with the obtained null model midpoint potentials, 144 ± 7 mV and −19 ± 4 mV. The high midpoint reduction potential is within the range usually obtained for other CYB561 proteins, but the low midpoint potential is somewhat lower than usual (see Table 1). Thus, the difference between the two reduction potentials is higher than that obtained usually for the other CYB561 proteins, although there have been publications with midpoint reduction potential differences as high as ~150 mV [4]. In earlier work this model, termed here as “null” was always considered [10,11,31,32,33].

Realistic model:

It can be shown that allowing any arbitrary 0 < α ≤ 1 the following midpoint reduction potential values would provide identical fit to the vs. matrix:
E_m,1_ = ^0^E_m,1_ − 60 log(1 + α)/NE_m,2_ = ^0^E_m,2_ + 60 log(1 + α)/NE_m,3_ = E_m,1_ + 60 log(α)/N(5)


Therefore, the true midpoint reduction potentials of the two hemes cannot be determined without further assumptions. Here we can consider another choice. (i) The midpoint reduction potential of either heme-*b* center is independent of the oxidation state of the other heme-*b* center (simple model) or (ii) each heme has two midpoint reduction potential values corresponding to the oxidation status of the other heme (coupled model). In the simple model the E_m,3_ = E_m,2_ restriction applies, and this results in a further constraint, allowing the determination of α and hence the true reduction potentials:(1 + α)^2^ = β α, with log(β) = (^0^E_m,1_ − ^0^E_m,2_)N/60(6)

With the accurate values from the fit in Figure 4B, ^0^E_m,1_ = 144.11 mV and ^0^E_m,2_ = −19.00 mV we obtain for the simple model β = 324, α = 3.1 × 10^−3^, E_m,1_ = E_m,4_ = 144.02 mV and E_m,2_ = E_m,3_ = −18.92 mV. Considering the coupled model, if we select α = 0.01, allowing ~1% “flow” through the bottom branch of the scheme, the obtained midpoint reduction potentials would be E_m,1_ = 143.83 mV, E_m,2_ = −18.761 mV, E_m,3_ = 13.857 mV and E_m,4_ = 111.21 mV. In this case the midpoint reduction potential of the low potential heme would be lowered from 13.857 mV to −18.761 mV as a result of the reduction of the high potential heme.

Electron paramagnetic resonance (EPR) spectra of oxidized and ~50% reduced Mm_CYB561D1 are shown in Figure 5 at temperatures 12.5 K and 20 K (panels A and B, respectively). These spectra are very similar to those obtained for the oxidized Hs_CYB561D2 at different temperatures [26] and, with one exception, the observed spectral peaks correspond well to previously assigned ones for other members of the di-heme cytochrome *b*561 protein family [30,32,34]. The characteristic g-values are marked and are shown with vertical lines. As expected, all spectral peaks are more intense at 12.5 K than at 20 K (which means a factor of 1.6 in kT between these temperatures). The peak at g_z_ = 4.30 has been assigned to non-heme iron and it is frequently observed in biological samples. The peak at g_z_ = 6.02 shows the presence of high spin ferric iron, and it has been associated with protein degradation. Protein degradation may be the result of the fact that the samples were stored at −80 ^o^C before the measurements. Despite its apparent intensity, this peak represents a minor component because of the spin-quantum-number-dependence of the EPR intensity [30]. The g_z_ = 4.30 and g_z_ = 6.02 peaks are of no interest for further considerations. However, the peak at g_z_ = 3.69 is rather prominent. At both temperatures, it loses most of its intensity but does not fully vanish upon ~50% reduction of Mm_CYB561D1. No significant EPR peak is present around g_z_ = 3.16, a signal usually assigned to rhombic heme environments in other cytochrome *b*561 proteins. A relatively broad and very weak peak might be present around g_z_ = 2.98, which vanishes or changes upon ~50% reduction, but this peak is not sufficiently clear. It has also been assigned earlier to protein degradation [30]. These observations suggest that in Mm_CYB561D1 both hemes have a highly asymmetric low-spin (HALS) character [9].

**Table 1 molecules-28-02261-t001:** Comparison of the properties of recombinant CYB561 proteins originally localized in different sources and expressed in different expression systems.

CYB561	Expression System	Localization	High-Affinity Asc Binding Site (mM)	Low-Affinity Asc Binding Site (mM)	Reduction Potentials (mV)		EPR Signal (*g* Value)		Ref.
					E_high_	E_low_	*g* _high-field_	*g* _low-field_	
Bt_CYB561A1	none	adrenal gland	?	?	150	60	3.69	3.13	[35]
Bt_CYB561A1	*E.coli*	adrenal gland	0.0053	0.394	171	81	3.72	3.15	[33,36]
Mm_CYB561A1	*S.cerevisiae*	adrenal gland	0.016	1.24	160	20	3.71	3.27	[27]
At_CYB561B1	*S.cerevisiae*	vacuolum	0.0054	0.336	165	57	3.63	3.16	[10,31],
Mm_CYB561D2	*S.cerevisiae*	?	0.268	7.3	141	43	3.61	2.96	[10,11]
Hs_CYB561D2	*P.pastoris*	?	?	?	109	26	3.75	3.65	[8,26]
Mm_CYB561D1	*S.cerevisiae*	?	0.045	2.34	144	−19	3.69	3.69	present

Our result supports also the unpublished observation by Asada et al. (see [26]) according to which study on the purified recombinant Hs_CYB561D2 expressed in *P. pastoris* cells, the EPR spectra might show two HALS-type heme signals and they overlapped around g_z_ = 3.6, but neither the rhombic EPR signal around g_z_ = 3.14, that would be indicative of a cytochrome *b*5-type alignment of the axial His ligands, nor a peak at g_z_ = 2.96 were recorded. Since the overlap of the two HALS-type heme signals is not unprecedented [9], it is likely that the difference in the reduction potential of the two hemes originates from structural differences relatively remote from the unpaired electron of the hemes.

## 3. Discussion

Spectral and physicochemical properties of the recombinant Mm_CYB561D1 are rather similar to those obtained earlier for other recombinant CYB561 proteins (Figure 1A and Table 1) and to those of the protein purified from bovine adrenal chromaffin granule vesicles [34]. There are, however, two minor differences; (a) the lower midpoint reduction potential of the recombinant Mm_CYB561D1 is below 0 mV and (b) only one EPR signal can be detected. Unfortunately, due to the lack of CD spectra of other CYB561 proteins we cannot compare our CD spectra to published ones. Our result suggests that there seems to be only minor electronic interaction between the two heme-*b* centers in the recombinant Mm_CYB561D1.

As demonstrated in Figure 3A,C, and as always observed for other CYB561 proteins [10,11,30,35,37,38], ascorbate cannot fully reduce both heme-*b* centers, whereas dithionite can. ASC at low concentrations first binds to the high affinity site. Full occupancy of the high affinity binding site can result in apparently 50% reduction of the protein by single electron donation. Depending on the midpoint reduction potentials of the two heme-*b* centers either exclusively one of them will be reduced or they will distribute the single electron. The second ASC binding site will be saturated at the highest concentrations. Apparently, the low potential heme cannot be completely reduced by the second electron due to the relative electron affinities of this heme and the ASC molecule docked in the low affinity binding site. In other words, an equilibrium is developed between the reduction of the low potential heme by ASC and the re-oxidation of the heme by the oxidized form of ascorbate. The spectral similarity of the two consecutive reduction products in case of Mm_CYB561D1 (Figure 3C) suggests that in this protein the two heme-*b* centers are in a very similar environment. Alternatively, it is also possible that in both steps the received electron is distributed almost equally between the two heme centers. This would then be different for certain other CYB561 proteins where there is a spectral difference between the consecutive reduction products.

Redox titration—similarly to other CYB561 proteins—also showed a bi-phasic pattern. This is readily explained by the presence of a low and a high potential heme (note that in this case no particular binding sites for the electron donor(s) are assumed). The null model (Equations (6), (7) and Figure 4B) can yield two definite values for the midpoint reduction potentials. However, the realistic model can yield the exact same fit as the null model with a whole range of possible midpoint reduction potential values. The assumption of no interaction yields two extreme potential values, while allowing a mutual effect of the redox state of either heme on the reduction potential of the other heme (coupling model) allows a whole range of less distant reduction potentials. For At_CYB561B1 it was possible to express and purify site directed double mutants with one pair of axial ligand His residues replaced, leaving only the low-potential heme in the protein [30]. The two measured reduction potentials were 178 and 20 mV for WT, whereas for the H83A/H156A and the H83L/H156L double mutants the reduction potential turned out to be 46 and 21 mV, respectively. Although the former, increased value (46 vs. 20 mV) may be explained by our realistic, coupled model, it appears that conformation alterations introduced by replacing the pair of His residues with different residues (e.g., Ala vs. Leu) and, consequently, the absence of the corresponding heme may also significantly affect the reduction potential of the remaining heme.

The putative structure of Mm_CYB561D1 was calculated by homology modeling (Figure 6). The structure highly overlaps with the two existing experimental structures. The docked ASC ligands as well as structural water molecules were also included in the modeling. The obtained Mm_CYB561D1 protein surface shows similar troughs to those in the crystal structures, serving as potential substrate binding sites. The overall pattern of the surface charge around the ASC binding sites on both sides of the protein are rather similar in the two experimental and the model structures (Appendix A). Interestingly, out of the three residues that contributed to hydrogen bonding the ASC on the cytoplasmic side, K77(79), K81(83) and R150(152) for At_CYB561B2 (Hs_CYB561A2), only R86, corresponding to K77, is involved in hydrogen bonding of ASC in our modelled structure. In place of K81 the model contains isoleucine (I90) and, although there is an arginine, R158, corresponding to R150 in At_CYB561B2, its side chain points in the wrong direction due to an amino acid insertion in the sequence between this position and the strictly conserved heme ligand, H166.

The rate coefficient of non-adiabatic electron transfer, according to the Marcus theory [39,40] is:(7)$$k=10^{13}T_{\mathrm{AD}}^{2}\exp\left(-\frac{{\left(\Delta G+\lambda \right)}^{2}}{4{\lambda k}_{B}T}\right){\;\sec}^{-1}$$
where k_B_ is Boltzman’s constant, T is absolute temperature, ΔG is the midpoint redox potential difference between the electron donor and acceptor pairs, λ is the reorganization energy and T_AD_ is the donor-acceptor electronic coupling term. T_AD_ is an exponential function of the distance (geometric distance or connectivity) between the donor and acceptor:(8)$$T_{\mathrm{AD}}=\exp\left(-½\beta \left(r-r_{0}\right)\right)\;\mathrm{or}\;\to \;T_{\mathrm{AD}}=\underset{i}{\prod }\varepsilon_{i}$$

In the first, packing density model, β = 0.9ρ + 2.8(1 − ρ), with ρ being the packing density of the medium spanning the space between the electron donor and acceptor and r_0_ is their contact distance, usually taken as 3.6 Å [41]. In the second, pathway model ε_i_ is the decay factor for the i^th^ step whose usual value is 0.6 for a covalent bond, 0.36 exp (−1.7(r − 2.8)) for a hydrogen bond, where r is the heteroatom distance in Å and 0.6 exp (−1.7(r − 1.4)) for a through space jump [42,43].

The most efficient electron transfer pathway from one ASC to the first heme, between the two hemes, and from the second heme to the other ASC was calculated by the program HARLEM. In these calculations, the aromatic rings of the hemes were taken as redox centers. As already pointed out by Ganasen et al. [6], there are no conserved amino acid residues and, therefore, no single best pathways in the two crystal structures between the hemes, and the best pathway in our model structure is also different from the others.

The program also yields estimates for the packing density, the distance decay constant and the maximal theoretical electron transfer rate between the redox centers. The parameters of the packing density model for the three structures are listed in Table 2. The parameters for the model structure of Mm_CYB561D1 are rather close to those calculated from the two crystal structures, corroborating the conclusion that it is the sufficient packing of the medium between the redox centers rather than any specific amino acid residues that determine the efficiency of electron transfer in the cytochrome *b*561 proteins.

## 4. Materials and Methods

### 4.1. Plasmid Construct, Yeast Transformation

The DNA fragments encoding the Mm_CYB561D1 (GenBank protein entry NP_001074789) and Mm_CYB561D1 fused to C-terminal thrombin cleavage site and His_6_ tag with the addition of 5′ and 3′ flanking *Bam*HI, *Eco*RI and *Not*I, *Sal*I restriction sites were codon-optimized for expression in *Saccharomyces cerevisiae* and chemically synthesized (GenScript, Piscataway NJ). The synthetic genes were inserted into pUC57 vector at *Eco*RV site and transformed into *E. coli* XL-1 Blue MRF’ competent cells. The plasmids were isolated with GenElute Plasmid Miniprep Kit (Sigma-Aldrich, St- Louis, MI). To clone the gene of recombinant Mm_CYB561D1 into expression vector the OWTCBD1CTH6 fragment was cut out from pUC59 with EcoRI/NotI digestion, isolated with GeneJET Gel Extraction Kit (Thermo Scientific, Waltham, MA) and ligated into the pESC-His plasmid (Agilent Technologies, La Jolla, CA) at EcoRI and NotI restriction sites, downstream of the GAL10 galactose-inducible promoter, to produce vector pMMCBD1H6G10. All recombinant plasmid constructs were verified by DNA sequencing.

The pESC-His expression vectors were transformed into *S. cerevisiae* cells (BMS1 overexpressing yTHCBMS1 strain [44]). For transformation yeast cells were grown in complex medium (YPAD broth, Agilent Technologies, La Jolla, CA). Competent cell preparation and transformation were performed according to manufacturer instructions (Agilent Technologies, La Jolla, CA). Transformed yeast cell lines were selected and maintained on synthetic dextrose minimal medium lacking histidine (SD-His).

### 4.2. Cell Growth and Membrane Preparation

Transformed yeast cells were grown in 250 mL portions in 1000 mL Erlenmeyer flasks in a temperature controlled incubator shaker at 150 rpm and 30 °C in growth medium described by Bonander et al. [44]. The final growth medium contained 20 mg/mL galactose as carbon source and 1 μg/mL doxycycline. Cell growth was terminated when the carbon source was used up completely.

Cells were harvested by low-speed centrifugation (at 4000 g_max_ and 10 °C for 10 min), washed twice in ice cold phosphate buffer (25 mM KH_2_PO_4_, 100 mM NaCl, pH 7), and suspended finally in ice cold homogenization buffer (50 mM MES-KOH, pH 7.0, 5 mM EDTA, 150 mM KCl, 200 mM sucrose) supplemented with 0.1% (*w*/*v*) Na-ASC and cysteine as well as with freshly prepared protease inhibitors. Cells were broken in a Bead Beater (Biospec Products, Bartlesville, OK, USA) with four 30 s cycles, with 2 min cooling intervals, using 0.5 mm glass beads. The unbroken cells and the cell debris were spun down (at 4000 g_max_ and 5 °C for 10 min). The yeast microsomal membrane fraction (YMMF) was obtained after high-speed centrifugation of the 4000 g_max_ supernatant (at 75,000 g_max_ and 4 °C for 60 min). The pellet was suspended in 50 mM phosphate buffer (pH 7.0) and either used immediately for solubilization or stored at −80 °C in the presence of 10% (*w*/*v*) glycerol until use.

### 4.3. Solubilization and Protein Purification

Solubilization of integral membrane proteins in the YMMF was performed in cold room (at ~6 °C) under continuous stirring (at ~150 rpm) for 90–120 min. Different detergents were tested (see the Results section) under slightly different experimental conditions. Since the dodecyl-β-D-maltoside (DDM) proved to be the best solubilizing agent among the tested ones, DDM was only used for solubilization throughout the further works. Insoluble material was pelleted by high-speed centrifugation (at 75,000 g_max_ and 4 °C for 60 min). The supernatant containing the DDM-solubilized proteins was concentrated by low-speed centrifugation using centrifugal filter units (Amicon Ultra-15 centrifugal filters with 50 kDa cut-off). Concentrated, DDM-solubilized proteins were stored in a deep-freezer (at ~−80 °C) in the solubilization buffer complemented with 10% (*w*/*v*) glycerol until use.

His_6_-tagged Mm_CYB561D1 was purified as detailed earlier [7] with little modifications. In this work (a) 0.5 mM DDM was present in all buffers after solubilization, (b) only 400 mM NaCl and 1% (*w*/*v*) glycerol was present in the affinity binding buffer, (c) 300 mM imidazole was present in the affinity elution buffer, and (d) Ni Sepharose High Performance resin (GE Healthcare Bio-Sciences AB, Uppsala, Sweden) was used as His_6_-tag binding resin. The affinity-purified protein was stored in deep-freezer (at ~−80 °C) until use in 50 mM phosphate buffer (pH 7.0), 0.5 mM DDM and 10% (*w*/*v*) glycerol.

Protein concentration in the YMMFs and the detergent-solubilized fractions were determined according to Markwell et al. [45] with BSA (Sigma, A4503) as protein standard. Total protein concentration in affinity purified fractions were not determined.

### 4.4. Optical Spectroscopy

All UV/Vis spectra were recorded with a Unicam UV4 spectrophotometer (in split-beam mode, in cuvettes with 1 cm optical path and at room temperature (at ~22 °C)). For calculation of the concentration of ascorbate- or dithionite-reduced CYB561 proteins the reduced-minus-oxidized difference spectrum and a differential molar extinction coefficient of ɛ(429–411 nm) = 222 mM^−1^cm^−1^ [46] was used.

Circular dichroism (CD) spectra were recorded with a Jasco J-815 spectropolarimeter in a 1 cm cuvette at room temperature, in the visible range for the alpha, beta bands (650–450 nm) and in the near UV range for the Soret band (475–350 nm). The Mm_CYB561D1 protein concentration for the visible range was 12 μM and for the Soret band 2.5 μM, and the spectra taken in the two regions were united after compensating for the concentration difference.

### 4.5. Redox Titration

Optical redox titration was carried out in 50 mM phosphate buffer, 0.5 mM DDM, ~1% (*w*/*v*) glycerol at room temperature under humidified Ar atmosphere and continuous stirring of solution in the cuvette. The sealed cuvette was equipped with an Ag/AgCl reference mini-electrode. Redox mediators used were potassium ferricyanide (+430 mV; 10 μM), 2,3,5,6-tetramethyl-p-phenylenediamine (+275 mV; 20 μM), trimethyl-hydroquinone (+115 mV; 20 μM), duroquinone (+5 mV; 20 μM), 2-hydroxy-1,4-naphthoquinone (−145 mV; 10 μM), and anthraquinone-2,6-disulfonic acid disodium salt (−225 mV; 10 μM). Reductive titration was performed by stepwise addition of 1 or 2 μL of sodium dithionite (stock solutions were 0.1 or 1 mM). Spectra between 360 and 660 nm were recorded after successive additions of the reductant and stabilization of the detected potential [47].

### 4.6. EPR Spectroscopy

Low temperature continuous wave EPR spectra, standard first harmonic in-phase signals with 100 kHz field modulation, were recorded on an X-band Bruker (Rheinstetten, Germany) EleXsys E580 EPR spectrometer equipped with a non-cryogenic helium cooling system. The protein solutions in 50 mM phosphate buffer, 0.5 mM DDM, ~1% (*w*/*v*) glycerol were prepared in the same way as for the redox titration measurement. The solutions were loaded into quartz tubes with an inner diameter of 3 mm and kept at 193 K prior to the measurement. The quartz tubes were loaded into the pre-cooled Super High Sensitivity waveguide probehead (Bruker) for measurement at 12.5 K and 20 K. The sample space was under vacuum during measurement and was flushed with helium gas when loading or exchanging samples, and the wave guide was flushed with nitrogen in order to remove atmospheric O_2_. All samples were measured with the following settings: microwave power, 9.46 mW; modulation amplitude, 10.0 Gauss; scan range, 0–3000 Gauss; conversion time, 15 msec; number of scans, 25. The gain was 60 dB (as recommended by Bruker). The microwave frequency was noted for each sample and was used to calculate the reported g-values. For baseline correction, the spectrum of the empty quartz tube was subtracted from that of the corresponding sample. The wide, feature-less baseline was fitted to a polynomial which was then subtracted from the spectra in order to remove the background but also preserve the signal-to-noise ratio. The spectral subtractions and fits were done using custom software written in Igor (WaveMetrics, Lake Oswego, USA).

### 4.7. Analysis of the Ascorbate and the Redox Titration Spectra

Difference absorption spectra relative to the fully oxidized sample’s absorption taken during titration were collected in a data matrix and subjected to Singular Value Decomposition (SVD) analysis [48]:**D** = **U S V**^T^(9)
where **D** is the data matrix containing the spectra, **U** and **V** are the matrices of the spectral and titration eigenvectors, respectively, and **S** is the (diagonal) matrix of the singular values. The number of significant components (i.e., the rank of the spectral matrix) was estimated based on the singular values, on the autocorrelation of the spectral and the titration eigenvectors and on the IND, REV and F tests [49]. The titration eigenvectors, i.e., the significant columns of the **V** matrix obtained with ASC titration, were fitted simultaneously to the equation
**V**_i_ = A_1,i_/(1 + K_1_/c) + A_2,i_/(1 + K_2_/c) + B_i_.(10)
where i = 1, 2, …, rank, c is the vector of the concentration values of the reducing agent (ascorbate), B_i_ is constant and K_1_ and K_2_ are the affinity constants of the two ascorbate binding sites at opposite sides of the membrane. The difference spectra, characteristic for the reduction of the protein through the successive titration steps, starting from the fully oxidized sample, can be calculated as
D_H_ = **U S** A_1_^T^, D_L_ = **U S** A_2_^T^(11)

The corresponding absolute spectra, presumably belonging to the high and low potential heme centers, are obtained by adding the absorption spectrum of the fully oxidized protein to the difference spectra D_H_ and D_L_, respectively.

Difference absorption spectra taken during Dth (redox) titration were similarly subjected to SVD analysis. The significant redox titration eigenvectors in the **V** matrix as a function of the measured reduction potential, E, in millivolts, were fitted simultaneously to the set of Nernst equations and yielded the midpoint reduction potentials of the heme-*b* centers. For the appropriate reaction scheme and the details of the analysis, see the Results section.

### 4.8. Structural Studies of the Mm_CYB561D1 Protein

The crystal structure of the *Arabidopsis thaliana* CYB561B2 (4O7G.pdb, [5]) was used as template in the calculation of the putative structure of the Mm_CYB561D1 by the program Modeller [50]. Internal water molecules, the two hemes and the two docked ascorbate molecules were included in the model. The level of homology between the two proteins is 51 identical, 45 highly homologous and 21 homologous amino acids out of a total of 229 in the sequence of CYB561D1. Before homology modelling various transmembrane domain predictor programs were used to find the putative six transmembrane segments of Mm_CYB561D1 (HMMTOP=CCTOP, TMHMM, Top Pred 1.10, TMPred, Predict Protein, MemBrain, OCTOPUS, Phobius, TOPCONS, Phlius). The alignment of the two sequences, as provided by Modeller agreed well with the predicted transmembrane segments, except for the first helix, where a 10 amino acid shift had to be manually introduced in the alignment suggested by Modeller to correctly model the first transmembrane helix.

Electron transfer pathway and parameter calculations were performed with the program HARLEM (https://crete.chem.cmu.edu/index.php/software/2-uncategorised/18-harlem, accessed on 27 February 2023) using the crystal structures and the homology structural model of Mm_CYB561D1.

## 5. Conclusions

We have presented absorption titration, CD, and EPR spectroscopic results as well as structural homology modeling calculations to support the idea that the recombinant Mm_CYB561D1 protein belongs to the ascorbate reducible, di-heme-*b* containing, transmembrane electron transporter cytochrome *b*561 protein family. The ascorbate reducibility and the redox properties of this protein have been analyzed in details by an improved, SVD based method of spectral analysis. Our analysis also took into account, for the first time, the possibility of coupling between the reduction potentials of the two hemes. We have shown that it is not possible to unequivocally determine the reduction potentials of the two hemes in these systems without further assumptions on the (lack of) interaction between the hemes. The minor differences between the characteristic biophysical parameters (midpoint reduction potential, EPR signal) for Mm_CYB561D1 and Mm_Cyb561D2 might point to their different biological function in cancer pathogenesis. The exact mechanism of tumor suppression activity of these proteins is still unknown and remains to be elucidated.

## Figures and Tables

**Figure 1 molecules-28-02261-f001:**
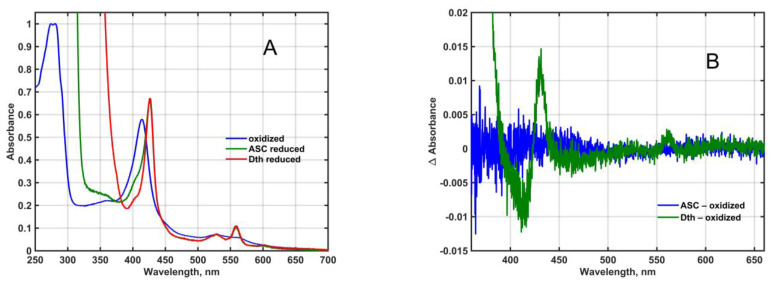
(**A**): Absorption spectra of affinity-purified, His_6_-tagged, recombinant Mm_CYB561D1 in 50 mM phosphate buffer (pH 7.0) in the presence of 0.5 mM DDM, normalized to the 280 nm absorbance for the oxidized form. (**B**): Ascorbate-reduced and dithionite-reduced minus oxidized difference spectra at pH 7 of the YMMF from yeast cells containing the empty expression vector only.

**Figure 2 molecules-28-02261-f002:**
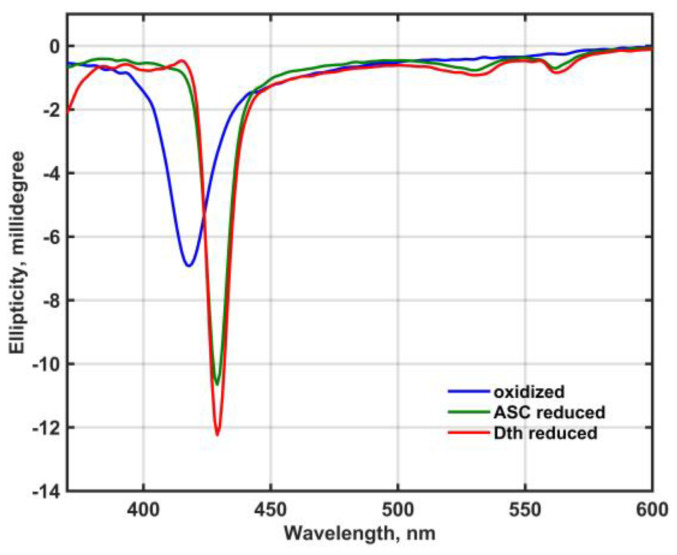
CD spectra of oxidized and reduced Mm_CYB561D1 in 50 mM phosphate buffer (pH 7.0) in the presence of 0.5 mM DDM. Spectra are combined from spectra taken at 2.5 μM and 12 μM Mm_CYB561D1 concentration in the near UV and the visible region, respectively.

**Figure 3 molecules-28-02261-f003:**
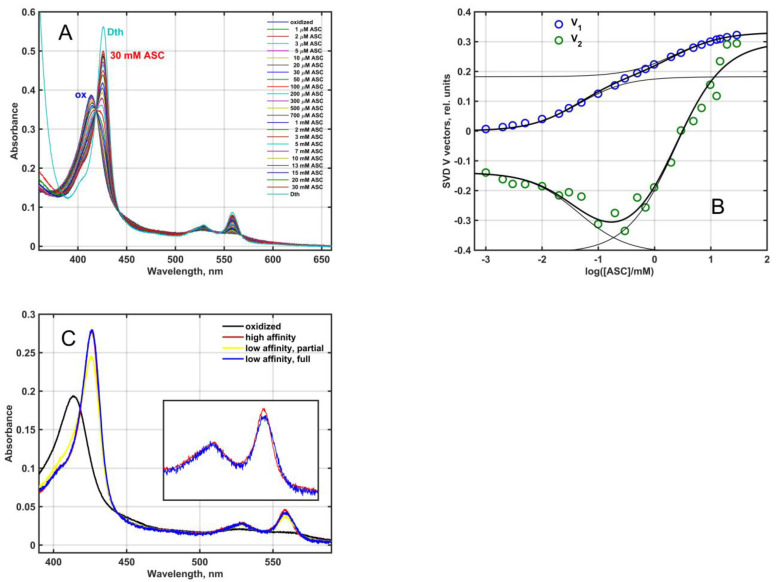
(**A**): Ascorbate dependent reduction of Mm_CYB561D1 in 50 mM phosphate buffer (pH 7.0) in the presence of 0.5 mM DDM. Mm_CYB561D1 concentration, 1.9 μM. A complete reduction could only be achieved by dithionite. (**B**): The two significant titration eigenvectors in the **V** matrix (circles) obtained from the SVD analysis of the difference spectral data matrix. Thick lines are fit to Equation (10) with thin lines showing the two individual saturation components. (**C**): Absolute spectra of the species reduced at low and high ASC concentrations (via high affinity and low affinity ASC binding), the extrapolated spectrum of the latter at full reduction and the spectrum of the fully oxidized protein.

**Figure 4 molecules-28-02261-f004:**
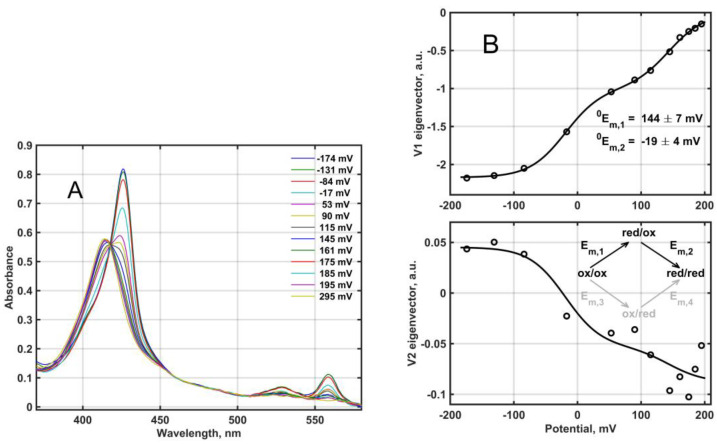
(**A**) Redox titration of Mm_CYB561D1 in 50 mM phosphate buffer (pH 7.0) in the presence of 0.5 mM DDM. Mm_CYB561D1 concentration, 2.7 μM. (**B**) The two significant titration eigenvectors in the **V**_S_ matrix (circles) obtained from the SVD analysis of the corresponding difference spectral data matrix. Lines are the fit to Equation (3) by the null model, Equation (4), corresponding to the upper branch (black) in the reaction scheme.

**Figure 5 molecules-28-02261-f005:**
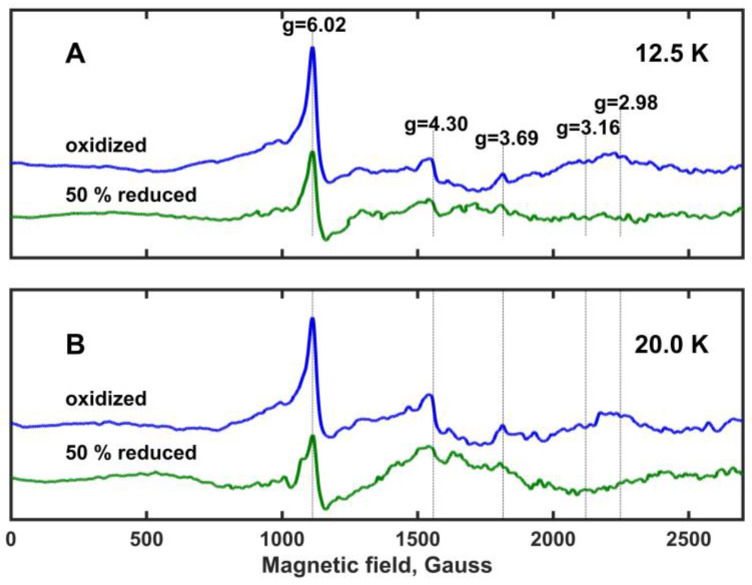
X-band EPR spectra of the oxidized and ~50% reduced Mm_CYB561D1 in frozen solution recorded at a temperature of (**A**) 12.5 K and (**B**) 20 K. Mm_CYB561D1 concentration, 1.9 μM. The spectra were recorded as a sum of 25 scans, with a microwave power of 9.46 mW and a modulation amplitude of 10.0 Gauss. The spectra are drawn to the same intensity scale. Known characteristic g-values for di-heme *b*561 cytochromes are indicated with vertical lines.

**Figure 6 molecules-28-02261-f006:**
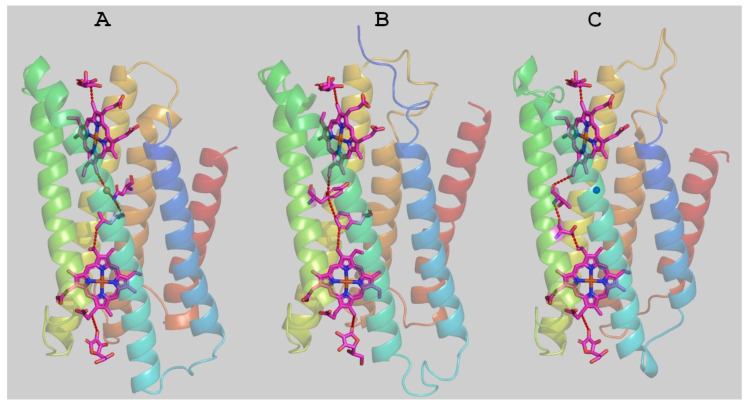
Experimental crystal structures of the *Arabidopsis thaliana* CYB561B2 (**A**) and the human duodenal CYB561A2 (**B**), as well as the homology modelled structure of Mm_CYB561D1 (**C**). The calculated best electron transfer pathways are seen with dashed red lines representing hydrogen bonds or through space jumps.

**Table 2 molecules-28-02261-t002:** Calculated parameters for the electron transfer between the two heme ring structures based on the packing density model.

	At_CYB561B2	Hs_CYB561A2	Mm_CYB561D1
Heme edge-to-edge distance, Å	15.6	15.5	15.6
Packing density, ρ	0.75	0.82	0.78
Decay constant, β, Å^−1^	1.37	1.25	1.33
Coupling term, T_AD_	2.21 × 10^−5^	6.10 × 10^−5^	3.34 × 10^−5^
Maximal ET rate, sec^−1^	4.87 × 10^4^	3.71 × 10^5^	1.12 × 10^5^

## Data Availability

The data presented in this study are available on request from the corresponding author.

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
