# Peer review of "Spectral and Redox Properties of a Recombinant Mouse Cytochrome *b*561 Protein Suggest Transmembrane Electron Transfer Function"

_molecules, 2023, doi:10.3390/molecules28052261_

Round 1
Reviewer 1 Report
This is an original work submitted by Bérczi et al. entitled: Spectral and redox properties of a recombinant mouse cyto-2 chrome b561 protein suggest transmembrane electron transfer function.
This manuscript is well-constructed, presenting enough results on the spectroscopic and redox properties of a recombinant mouse cyto-2 chrome b561 protein. However, I have some comments and suggestions to be addressed.
Major revisions are suggested before the publication of the manuscript.
There are contradictions between the results, the analysis and the conclusions. For example, the EPR results indicates the same signals for the two heme groups (g=3.69), which accounts for a very similar chemical environment, however, the redox potentials determined are abnormally different with respect to others previously reported.
It is not well understood why the EPR signals for degradated-protein iron (g=6.2) and non-heme iron (g=4.3) are much more intense than those for the heme irons (g=3.69). Please compare with the epr spectra of reference 33 of yourmanuscript.
In the experimental section the concentration of the protein used for each experiment is mentioned, however, it would be worthwhile to include this concentration in the figure captions of each result.
It is not mentioned whether the EPR studies were performed with fresh samples of the protein or with samples stored after some time at -80 degrees. This is relevant as other studies have reported the dependence of EPR signals on storage time (DOI 10.1016/j.biochi.2012.02.021 ).
If the authors have the possibility to do so, it would be worthwhile to perform the determination of redox potentials with a spectroelectrochemical assay using EPR coupled to potentiometry, e.g. DOI 10.1016/0167-4838(85)90196-7 and DOI 10.1021/bi9015007.
Line 429 The method used for spectroelectrochemistry appears to be that reported by Dutton although using a conventional cell and a microelectrode. It would be good to cite DOI 10.1016/S0076-6879(78)54026-3.
Line 257 Place the EPR experimental conditions (gain, power, temperature, etc.) in the figure 5 caption
The authors should elaborate more on the conclusions.
Author Response
We are grateful for the reviewer’s critical reading of the manuscript and for the comments and suggestions that helped us to improve the paper. Our responses to the points raised by the Reviewer are as follows:
There are contradictions between the results, the analysis and the conclusions. For example, the EPR results indicates the same signals for the two heme groups (g=3.69), which accounts for a very similar chemical environment, however, the redox potentials determined are abnormally different with respect to others previously reported.
Response: Our observation is that in Mm_CYB561D1 the two heme cofactors (and chromophores) are similar in two aspects: (i) their EPR signals overlap at g=3.69 and (ii) their absorption (absorption difference) spectra are also very similar in the visible range. There is at least one published example of similar EPR overlap (ref 9 in the ms, Recuenco et al. 2013). Nevertheless, rather independent of the g values, in most known examples the midpoint reduction potential difference is in the 100 mV range. In our present case the high potential is in the usual range, only the low potential is somewhat lower than the previously published values. Accordingly, we have re-phrased the corresponding sentences in the revised manuscript (lines 195-200). In our opinion there should not necessarily be direct correlation between the EPR results and the midpoint reduction potentials. The EPR signal reports primarily about the degree of localization of the unpaired electron orbital, whereas the redox potential is expected to depend more on the electrostatic, polar or hydrogen bonding environment of the heme.
We have added a sentence at line 269-271 to reflect on the apparent contradiction the Reviewer has called our attention to.
It is not well understood why the EPR signals for degradated-protein iron (g=6.2) and non-heme iron (g=4.3) are much more intense than those for the heme irons (g=3.69). Please compare with the epr spectra of reference 33 of yourmanuscript.
Response: There must be differences in the sample purity and the pre-measurement history of the samples between ref. 33 (Liu et al. 2007) and our work. Liu et al. studied chromaffin granule cytochrome b561 obtained from a bacterial expression system, whereas our Mm_CYB561D1 has been expressed in and purified from yeast. EPR measurement was done at 8K in ref. 33 and at 12.5 and 20 K in our case. It is known that the signal intensity has high temperature dependence; the lower the temperature the higher the signal intensity for the heme at g=3.6 – 3.7 (ref. 9 in the ms., Recuenco et al. 2013). In another example (ref. 32 in the ms., Kamensky et al. 2007) EPR measurements had been done on chromaffin granule CYB561, and the g=6.2 and g=4.3 signals, relative to the heme signals, depend strongly on the purification degree of the sample. Nevertheless, as also pointed out by the Reviewer, storage of the sample at – 80 oC may contribute to protein degradation, and this is now explicitly stated in a new sentence at line 244-245. It is still reassuring that the ascorbate titration and the redox titration measurements were performed on similar samples and the overall intactness of the protein sample was verified.
In the experimental section the concentration of the protein used for each experiment is mentioned, however, it would be worthwhile to include this concentration in the figure captions of each result.
Response: These data are now included in the figure captions, except for Figure 1, where the spectra were normalized, so the absolute spectral amplitudes are arbitrary.
It is not mentioned whether the EPR studies were performed with fresh samples of the protein or with samples stored after some time at -80 degrees. This is relevant as other studies have reported the dependence of EPR signals on storage time (DOI 10.1016/j.biochi.2012.02.021 ).
Response: It is now mentioned in a new sentence, line 244-245, also, see above.
If the authors have the possibility to do so, it would be worthwhile to perform the determination of redox potentials with a spectroelectrochemical assay using EPR coupled to potentiometry, e.g. DOI 10.1016/0167-4838(85)90196-7 and DOI 10.1021/bi9015007.
Response: This could be an elegant experiment and also the next logical step but, unfortunately, we do not have the necessary technical background. This would also require a significantly improvement of the purification yield and the cost-efficiency of the proteins’s production.
Line 429 The method used for spectroelectrochemistry appears to be that reported by Dutton although using a conventional cell and a microelectrode. It would be good to cite DOI 10.1016/S0076-6879(78)54026-3.
Response: The Dutton paper is now cited.
Line 257 Place the EPR experimental conditions (gain, power, temperature, etc.) in the figure 5 caption
Response: We have added the most relevant details of the spectrometer settings in the legend of Fig. 5. The gain has been added to the corresponding methods section (line 477).
The authors should elaborate more on the conclusions.
Response: We have added sentences to both the Discussion and the Conclusions sections, to further emphasize the message of our manuscript: lines 309-312, 542-545, 547-549.

Reviewer 2 Report
The work by Bérczi et al reports results of isolation of partially purified cytochrome b561 and its characterization using UV-vis, CD spectroscopies and EPR. Authors report that cytochrome b561 contains two heme sites possessing different redox properties, which were supported using redox titration experiments. Both heme sites are capable of reacting with ascorbate, whereas some other cytochrome species do not react with ascorbate and can be reduced to ferrous state by dithionite. In general, the work in suitable to publication in Molecules, however, attention should be paid to the following point.
Complete reduction of both ferric hemes to ferrous ones requires large excess ascorbate that was explained by the presence of two ascorbate-binding sites possessing different affinity toward substrate. Reaction of metal centers with ascorbic acid can reversible due to quite pronounced oxidizing properties of ascorbyl radical and dehydroascorbate, and these observations can be explained by the reversibility of the process. Thus, possible reaction between Fe(II) cytochrome b561 and dehydroascorbate should be examined.
Author Response
We are grateful for the positive opinion of the Reviewer and for the thoughtful comment that we address below:
Complete reduction of both ferric hemes to ferrous ones requires large excess ascorbate that was explained by the presence of two ascorbate-binding sites possessing different affinity toward substrate. Reaction of metal centers with ascorbic acid can reversible due to quite pronounced oxidizing properties of ascorbyl radical and dehydroascorbate, and these observations can be explained by the reversibility of the process. Thus, possible reaction between Fe(II) cytochrome b561 and dehydroascorbate should be examined.
Response: The existence of the two binding sites on the opposite water-exposed surfaces of the protein has been proven by numerous studies and, more recently, by the two X-ray structures. We are the first to show this by ascorbate titration experiments for Mm_CYB561D1. We hypothesized that the lack of complete protein reduction by even the highest applied ascorbate concentration is due to the comparable electron affinity of ASC and the (low potential) heme. In other words this is exactly what the Reviewer suggests, so that the re-oxidation of the protein by the oxidized ascorbate must be taken into account. To clarify this point we added a new sentence to the manuscript at line 296-298.
Round 2
Reviewer 1 Report
After reviewing this second version of the article, I am very satisfied with the enrichment of the discussion and conclusions. In my opinion, the article will represent an important contribution to the understanding of the reaction mechanism of Cytochrome b561 through the determination of their redox properties and spectroscopic characterization. mechanisms of action of synthetic peptides in antimicrobial activity.
Reviewer 2 Report
I recommend acceptance of the manuscript in current form